# The Influence of Alcohol Consumption, Depressive Symptoms and Sleep Duration on Cognition: Results from the China Health and Retirement Longitudinal Study

**DOI:** 10.3390/ijerph191912574

**Published:** 2022-10-01

**Authors:** Tianyue Guan, Chao Zhang, Xuanmin Zou, Chen Chen, Li Zhou, Xiaochang Wu, Jiahu Hao

**Affiliations:** 1Maternal, Child, and Adolescent Health Department, Anhui Medical University, Hefei 230032, China; 2MOE Key Laboratory of Population Health Across Life Cycle, Hefei 230032, China; 3Anhui Provincial Key Laboratory of Population Health and Aristogenics, Anhui Medical University, Hefei 230032, China

**Keywords:** alcohol, depressive symptoms, sleep duration, middle-aged and elderly Chinese adults, a national survey

## Abstract

Aim: Healthy cognition-related factors include alcohol consumption, depressive symptoms, and sleep duration. However, less is known about the role of these factors in the dyad or tripartite relationships. In this study, we examined whether there were potential mediation effects, moderation effects, and interactions between these factors in the longitudinal study. Methods: Both cross-sectional data analysis and a longitudinal study were performed using baseline and 2018 data from the China Health and Retirement Longitudinal Study (CHARLS) cohort. CHARLS is a nationwide survey program covering 450 villages and 150 counties in 28 provinces that aims to investigate comprehensive demographic information. After selecting participants from the CHARLS cohort, 15,414 were included in the study. Non-drinkers, those who drink more than once a month, and those who drink less than once a month were defined by their alcohol consumption. Depressive symptoms were defined as nondepressed (less than or equal to 12) and depressed (more than 12). Sleep duration was defined as 7–8 h per night, ≤6 h per night, and ≥9 h per night. The total cognitive scores were calculated from memory, orientation, and executive tests. The PROCESS macro in SPSS was used to analyze all mediations and moderating mediations. Results: Alcohol consumption has a positive correlation with cognition. The global cognition z scores of participants with depressive symptoms were significantly lower than those of the control (all *p*’s < 0.001), in different models. The memory score (β: −0.148; 95% CI: −0.240 to −0.056; *p* = 0.002), the executive score (β: −0.082; 95% CI: −0.157 to −0.006; *p* = 0.033), and the global cognition score (β: −0.105; 95% CI: −0.187 to −0.023; *p* = 0.012) of participants defined as ≤6 h per night were, obviously, less than the control (7–8 h per night). An association between depression and alcohol consumption has been found, and the protective effects have been reversed by depression, which caused the cognition decline. Sleep duration was identified as a moderator, influencing the relationship between depressive symptoms and cognitive function. Besides, there was an interaction causing cognition decline among alcohol consumption, depressive symptoms, and sleep duration. Cognitive function showed a marked downward trend with increasing age. Conclusions: In this model, depression primarily mediates the relationship between alcohol consumption and cognition, and sleep duration changes the mediation effect. Furthermore, there is a significant interaction between alcohol consumption, depressive symptoms, and sleep duration, which are significantly associated with cognitive function.

## 1. Introduction

Increasing brain atrophy is associated with increasing age, with the elderly being forced to suffer from cognitive decline and even neurocognitive disorders. As an intermediate stage between normal cognition and dementia, mild cognitive impairment (MCI) may be treated clinically by therapeutic interventions, which impose a substantial economic burden on our aging society as the global population ages [1,2]. Additionally, basic demographic characteristics (like gender) [3] and lifestyle (such as smoking) [4] were considered risk factors associated with cognitive function.

Alcohol consumption has been a major public health problem, and the significant health effects cannot be ignored, even at low levels of alcohol consumption [5,6,7]. Notably, a recent study indicated that the minimum health hazard dose of alcohol consumption is none [8]. Rumgay et al. [9] further pointed out that more than 740,000 people suffered from cancer due to alcohol exposure in 2020, and increasing cancer morbidity is related to the consumption of alcohol, even in small amounts. There is controversy regarding the risks of moderate and low levels of alcohol consumption, while heavy drinking is believed to be a risk factor for neurocognitive disorders. Some studies pointed to the protective effects of moderate alcohol consumption on cognitive function [10,11]. Meanwhile, other studies also suggested that alcohol exposure may cause cognition impairment [12,13]. Cognitive function broadly refers to multiple dissociable, including visuospatial function, memory, executive function, orientation, etc. [14,15]. These domains of cognitive function may be affected by alcohol consumption, but this is unclear. The mechanisms have not yet been elucidated, and the mediation path has not been validated for alcohol consumption.

The relationship between sleep duration and cognitive function was controversial. Studies have found an association between sleep duration and cognitive function [16]. The authors of another study found an inverted U-shaped association between sleep duration and cognition, based on statistical analysis [17,18,19]. Both too little and too much sleep can cause cognitive decline. However, research has shown that sleep duration did not correlate significantly with cognition [20]. Longitudinal studies show that patients with depressive symptoms in early or middle life may have cognitive decline, cognitive impairment, or dementia [21,22]. When people have depressive symptoms, their memory, executive function, visuospatial function, and linguistic abilities are diminished [23]. Nevertheless, some researchers reported that depressive symptoms exhibit a significant protective effect on cognitive function in MCI patients, by causing social withdrawal [24,25].

According to a study, sleep duration and depression symptoms interact with cognition in the elderly [26]. There may be pathways from sleep disturbances and depression to cognitive functions, so the link between depressive symptoms and sleep duration cannot be ignored [27,28]. Besides, alcohol consumption was associated with major depressive disorder [29]. Despite evidence that alcohol consumption, depression, and sleep duration are associated with cognitive function, most studies have tested these factors individually or assessed a single domain of physical functioning. Whether there are mediation paths and interactions for cognition among alcohol consumption and depressive symptoms and sleep duration has, so far, not been verified.

We aim to (1) test whether alcohol consumption, depressive symptoms, and sleep duration are significantly related to cognitive function at the baseline in the longitudinal study; and (2) examine whether there are possible mediation effects, moderating effects, and interactions among alcohol consumption, depressive symptoms, sleep duration, and cognition in the longitudinal study. We hypothesize that depressive symptoms and sleep duration play a moderating and/or mediating role between alcohol consumption and cognitive function, and/or there are interactions among them. Together, our study provides new evidence for whether alcohol exposure can cause cognitive decline by mediating depressive symptoms and/or interacting with sleep duration.

## 2. Materials and Methods

### 2.1. Data

Cross-sectional data analysis and longitudinal study were performed using baseline and 2018 data from the CHARLS cohort. CHARLS is a community-based cohort of Chinese residents aged 45 years and older and their spouses, collected mainly through interviews [30]. The baseline survey was performed in 2011, and four waves of data have been collected to date for the cohort (2011–2013–2015–2018). Ethical approval for all the CHARLS waves was granted from the Institutional Review Board at Peking University, and all participants in the cohort provided written informed consent. The IRB approval number for the main household survey, including anthropometrics, is IRB00001052-11015; the IRB approval number for biomarker collection is IRB00001052-11014. This study followed the Strengthening the Reporting of Observational Studies in Epidemiology (STROBE) reporting guideline.

After fitting baseline data with 2018 tracking data, 17,596 participants in the cohort were eligible for inclusion. Since the primary outcome for this study was cognitive function, 1231 participants who did not meet the criteria were excluded due to: (1) brain damage/mental retardation (n = 523); (2) memory impairment, such as Alzheimer’s disease, brain atrophy, and Parkinson’s disease (n = 277); (3) cancer or malignancy other than minor skin cancer (n = 180); and (4) emotional, nervous system, or psychiatric problems (n = 251). After excluding participants for the criteria detailed above, participants with incomplete baseline information were excluded (n = 951), and data were available for 15,414 participants. The complete participants’ selection and classification process are shown in Figure 1. The participants were divided into 8 groups according to sleep duration (≤6 h, 7–8 h, and ≥9 h per night), depressive symptoms (nondepressed and depressed), and alcohol consumption (non-drinkers, more than 1 drink per month, and less than 1 drink per month), respectively, for the analyses. The complete participants’ selection and classification process are shown in Figure 1.

### 2.2. Outcome

Cognitive assessments were performed on all waves of the CHARLS cohort. Cognitive ability is assessed through cognitive scores, including ratings of memory, execution, and orientation. Memory assessments consisted of two components, immediate recall and deferred recall, to evaluate episodic memory function; the result was the sum of words between 0 and 20. The orientation consists of four questions: day of the week, month, month, and year (range 0–4). The test consists of two parts, subtracting 100 from 7 consecutively (up to 5 times) and duplicating the intersecting pentagons, and the executive score was the sum of two parts ranging from 0 to 8. The global cognition scores were based on the sum of the three domains (score range: 0–32). The z scores of three tests and the global cognition were generated by standardizing comparison across tests, which has been commonly used to obtain z-scores for global cognitive function [31]. Baseline cognition scores were used as outcomes in the cross-sectional study, and cognition scores in the follow-up were used in the longitudinal study. Furthermore, the reliability and validity of the above tests has been well-proven [32,33].

### 2.3. Exposure

Alcohol consumption. Drinking frequency was divided into three categories in the baseline investigation. Alcohol consumption was accessed through the following questions: “Did you drink any alcoholic beverages, such as beer, wine, or liquor in the past year? How often?” Past research demonstrated that the relationship between alcohol intake frequency and cognition of the elderly was found to be significant [34,35]. The control group was defined as non-drinkers, and the exposed group was defined as having more than one drink and less than one drink per month.

### 2.4. Mediator

Depressive symptoms. Depressive symptoms were measured using the Center for Epidemiological Research (CES-D) Depression Scale and validated in the Chinese elderly [36]. This is a 10-item version, each rated on a 4-point scale from 0 (almost or never) to 3 (almost or always). Total scores for the CES-D elements range from 0 to 30, with higher scores meaning higher levels of depressive symptoms. Previous research has shown that scores of 12 and above are used to define depression [36,37]. The control group was nondepressed (less than or equal to 12), and the exposure group was depressed (more than 12). The reliability and validity of this method have been verified through 742 participants of CHARLS aged 60 and older [37].

### 2.5. Moderator

Sleep duration. Initial sleep time was self-reported, and there was no category for the face-to-face interview. The control group was defined as 7–8 h per night, and the exposure group was defined as ≤6 h per night and ≥9 h per night. This conformed to recommendations of the American Academy of Sleep Medicine [38].

### 2.6. Covariates

Structured questionnaires were used to collect basic demographic, lifestyle, and medical history data, including age (years), gender, education, partner or not, systolic blood pressure (mmHg), diastolic blood pressure (mmHg), BMI (weight (kg) divided by height (m) squared), smoking, self-reported hypertension, diabetes, dyslipidemia, coronary artery disease, stroke, chronic lung disease, asthma, etc. Education level was divided into five categories: no formal education or illiteracy, no primary school diploma, secondary school diploma, university, postgraduate education, or other. Cigarette consumption was defined as nonsmokers and smokers. The blood pressure of each participant, which was divided into systolic and diastolic, was measured and averaged three times according to the standard protocol. Accumulating evidence suggests that hypertension and diabetes are important risk factors for neurocognitive disorders and dementia [39,40]. Hypertension was defined as hypertension reported by patients receiving antihypertensive medication. Diabetes was defined as patient-reported use of diabetes treatment and/or antidiabetic medicines.

### 2.7. Statistical Analysis

In this study, the outcome was treated as a quantitative variable, and the exposure, mediator, and moderator were assessed as categorical variables. All descriptive variables were expressed as number (%) and continuous variables as mean (SD). This study design included cross-sectional and longitudinal studies.

Cross-sectional outcomes were assessed using stratification analysis and generalized linear models. The chi-squared test was used for categorical variables, and Student’s *t*-test was used for continuous variables. A stratified analysis approach was used to describe participants’ demographics, grouped by alcohol consumption, depressive symptoms, and sleep duration, and to verify differences between groups. Generalized linear models assessed associations with alcohol consumption, depressive symptoms, sleep duration, and cognitive ability. We designed different models by adjusting for different covariates.

The multiple cognitive measures were acquired at baseline and 2018 follow-up, constituting repeated-measures data. After the z scores at baseline and follow-up were fitted, we used a PIN to detect repeated measures of cognitive function and deleted the missing data. We examined adjusted longitudinal associations and interactions between alcohol consumption, depressive symptoms, sleep duration, and cognitive performance using linear regression models with generalized estimating equations (GEE). Trajectories of cognitive decline with increasing age were examined using the same models.

We tested the indirect association of depression as a mediator, when the z scores at the 2018 follow-up were used as the outcome, to explain the association of alcohol consumption with cognitive function (Figure 2). The total effect, natural indirect effect (NIE), and natural direct effect (NDE) of alcohol consumption on cognition were estimated using SPSS PROCESS v3.3 (Model 4). Furthermore, we used SPSS PROCESS v3.3 (Model 59) to test for indirect effects of depressive symptoms and the moderating effect of sleep duration under different models. There were missing values for variables included in the regression analysis.

Finally, we undertook sensitivity analyses around depressive symptoms for both the cross-sectional and longitudinal analyses. Two-sided *p* values < 0.05 were considered statistically significant. Statistical analyses were performed by SPSS 21 (SPSS for Windows, Chicago, IL, USA) and R (v4.1.3).

## 3. Results

### 3.1. Baseline Characteristics of the Participants

The baseline demographics of the study population are shown in Table 1. Participants defined as nondrinkers (66.9%), nondepressed (66.9%), and sleeping ≤6 h per night (49.6%) had the highest proportions. The average age of all participants was 59 years, and participants defined as having finished a middle school education had the highest proportion (27.2%). Overall, 80.1% of participants lived with their partners. Hypertension was a common condition, with the highest prevalence of 24.1%, compared with other common diseases.

### 3.2. Baseline Alcohol Consumption, Depressive Symptoms, Sleep Duration, and Cognition z Scores in Cross-Sectional Analyses

As shown in Table 2, alcohol consumption, depressive symptoms, and sleep duration were statistically correlated with cognitive function. The positive associations between drinking more than once per month, as well as memory score, orientation score, executive score, and global cognition score, were all verified statistically (all *p*’s < 0.05). Global cognition z scores of participants with depressive symptoms were significantly lower than those of the control (all *p*’s < 0.001) in Model 1 and Model 2. Different sleep duration group comparisons showed that the memory score (β: −0.148; 95% CI: −0.240 to −0.056; *p* = 0.002), the executive score (β: −0.082; 95% CI: −0.157 to −0.006; *p* = 0.033) and the global cognition score (β: −0.105; 95% CI: −0.187 to −0.023; *p* = 0.012) of participants defined as sleeping ≤6 h per night obviously less than the control (7–8 h per night). Similarly, there was a positive relationship between education level and cognition z scores (all *p*’s < 0.05). In Model 2, the orientation score (β: −0.003; 95% CI: −0.005 to −0.001; *p* = 0.018) and executive score (β: −0.003; 95% CI: −0.005 to 0.000; *p* = 0.022) were both significantly negatively correlated with diastolic. The two groups were positively associated with orientation scores (β: 0.043; 95% CI: 0.000–0.086; *p* = 0.048), total cognitive scores (β: 0.048; 95% CI: 0.004–0.092; *p* = 0.034), and cigarette smoking consumption.

### 3.3. Mean Difference in Overall Rate of Change in Cognitive Decline during Follow-Up

The longitudinal associations and interactions between alcohol consumption, depressive symptoms, sleep duration, and cognition are shown in Table 3. After adjusting the confounding variables, the global cognition score of participants defined as drinking more than once per month was statistically 0.156 points higher than in the control group (β: 0.156; 95% CI: 0.007 to 0.305; *p* < 0.05). Similarly, the global cognition score of participants defined as depressed was statistically 0.271 points lower than that of in the control group defined as nondepressed (β: −0.27; 95% CI: −0.411 to −0.130; *p* < 0.001). Sleep duration, defined as 6 h or less per night (β: −0.056; *p* < 0.001), was significantly negatively associated with total cognitive scores compared with the controls (7–8 h per night), while sleep duration defined as ≥9 h per night (β: 0.025; *p* < 0.001) was positively associated with the global cognitive score. The longitudinal study showed a statistical relation between gender and cognitive function (β: 0.034; 95% CI: 0.002 to 0.067, *p* < 0.05). We further examined the statistical interaction between these variables; the results show significant interaction between alcohol consumption, depressive symptoms, and sleep duration (*p* < 0.05).

Figure 3 shows the process of cognitive decline with age. As illustrated in Figure 3, all four cognition z scores showed a marked downward trend with increasing age, after adjusting for covariates. The curves of cognition showed a downward trend from age 50 to 55 years, while the curves shifted to an upward trend, when the sampling area reached an inflection point, and then cognition decline persisted.

### 3.4. Adjusted Direct and Indirect Associations Model and Conditional Indirect Effect Model during Follow-Up

Table 4 shows the overall natural direct effect (NDE) and natural indirect effect (NIE) of alcohol consumption and depressive symptoms on cognition during follow-up. In Model 3, the indirect effect suggested that the global cognition score of participants defined as having alcohol exposure was 0.003 points lower than that of the control group (β: −0.003; 95% CI: −0.005 to −0.001). The proportion mediated was 100%, which was fully mediated by depressive symptoms. The results of different models were roughly similar. The effects of alcohol consumption on cognitive function are primarily mediated by causing depression, while there is no direct correlation between alcohol exposure and global cognition.

As shown in Table 5, sleep duration was included in the model as a moderator. The results of the different models proved that the moderated mediation model was valid. Model 59 in the SPSS process was used to access the moderated mediation. In Model 3, there was a significant mediating role of depressive symptoms at a low level (effect: −0.004, 95% CI: −0.008 to −0.001) and the mean level (effect: −0.003, 95% CI: −0.005 to −0.001), while there was no significant mediating role at a high level (effect: −0.002, 95% CI: −0.005 to 0.001). Thus, it can be seen that the moderated mediation model was valid because the mediation via depressive symptoms was different at different levels. The moderated mediation model is illustrated in Figure 4.

### 3.5. Sensitivity Analyses

The cross-sectional and longitudinal data were reanalyzed by excluding all participants with depressive symptoms, to exclude the possible effects of depressive symptoms on the outcomes. This analysis yielded similar results to the primary analysis. The declining trajectory of cognitive function with age was also similar to the primary analysis, showing a gradual downward trend. However, some correlations, especially longitudinal analyses, were no longer statistically significant across domains (Appendix A).

## 4. Discussion

This study indicates that alcohol consumption is positively associated with cognition, and participants defined as depressed and sleeping ≤6 h per night show a greater decline in cognition. After adjusting for covariates, depressive symptoms mediated the correlation between alcohol consumption and cognition and reversed their protective effects, which caused cognitive decline. The impact of depressive symptoms on cognition was influenced by sleep duration. Besides, there was an interaction causing cognition decline among alcohol consumption, depressive symptoms, and sleep duration. Cognitive function showed a marked downward trend with increasing age.

There has been much discussion and controversy about the association between moderate alcohol consumption and cognition. A survey of a prospective cohort study in the USA indicated that light–moderate (less than 8 drinks per week for women and less than 15 drinks per week for men) alcohol intake was positively related to better cognitive performance in the middle-aged and older [41]. However, the latest research from Oxford University suggested that any alcohol exposure was harmful to the brain, and deteriorating brain conditions have been linked to increased alcohol consumption [42]. Both sleep duration and depression have been reported to be positively related to cognition decline [17,28,43,44], which was similar to this present study. Although all these factors are considered to be risk factors for cognition, this is the first study to identify the mechanism of depressive symptoms mediating the relationship between alcohol consumption and cognition when sleep plays a moderating role. There was an apparent interaction between cognition and alcohol consumption, depression, and sleep duration (*p* < 0.05). Meanwhile, previous studies suggested that cognition may be mediated by depressive symptoms and/or sleep duration. An epidemiological study of 246 nondemented elderly adults in Shenzhen, Guangdong Province, China, reported that depressive symptoms may mediate the association between sleep disorder and cognition decline [45]. Furthermore, another cross-sectional study investigated 204 MCI elderly adults living in the community, indicating that about 26% of the cognitive impact of depressive symptoms was mediated by sleep quality [27].

Importantly, our study indicated 100% of the effects of alcohol consumption on cognitive function were mediated through depressive symptoms, while Preacher et al. [46] suggested the fact that the proportion mediated was 100% did not mean only one mediator, which was related to sample size. Poor sleep accelerates the direct and indirect association between cognitive decline and alcohol consumption, which was mediated by depressive symptoms. This provides new insight into the mechanism of cognition decline caused by alcohol exposure in the elderly. Alcohol directly contributes to depressive symptoms and risk factors for cognitive decline, but it does not affect cognitive function directly. Due to the moderating role, the direct and indirect effects of depression can be accelerated or modified by sleep duration. A study published in 2022, using data from UK Biobank and investigating 371,463 adults, reported that the lifestyle of participants with light–moderate alcohol consumption, with a higher vegetable intake, with higher physical activity levels, and with less smoking, was healthier than that of abstainers [47]. Therefore, the observed benefits of light–moderate alcohol consumption were associated with multiple kinds of lifestyles, and there was no health benefit for alcohol exposure, after excluding this factor. The conclusions of our study are approximately consistent with this study. There were potential moderators such as sleep duration affecting and/or reversing the final results for the mechanism by which alcohol affected cognitive function. However, moderate alcohol consumption had protective effects on cognition in the cross-sectional study. This can be verified by sensitivity analyses. We found no statistical association between alcohol consumption (> = one time per month) and cognitive function in a longitudinal analysis, after excluding participants defined as depressed. Enough sleep (7–8 h for adults) may contribute to a decreased risk of cognitive impairment caused by depressive symptoms. These findings provide valuable evidence for the development of policies and guidelines on alcohol consumption among the Chinese population and suggest policy implications for public health.

## 5. Strengths and Limitations

The greatest contribution of this study was the new theory that depression mediated the link between alcohol consumption and cognition, when sleep duration plays a moderating role. We not only tested the association between these factors and cognitive function using cross-sectional analyses but also fit both the baseline and follow-up data to verify the correlation. In addition, we used follow-up cognition scores to draw firm conclusions about the association in the moderated mediation model. The large sample size also provided good statistical power for association analysis. We examined prior alcohol consumption, depressive symptoms, and sleep duration as independent risk factors for cognition and described the trend of cognition decline with age. More importantly, we further proposed a model by which alcohol exposure affected cognition. However, to date, there is no direct proof that the model is valid, which requires further in-depth investigation in the future to reduce the cognitive impairment of alcohol consumption. Thus, our findings have important implications for practice and policy. They may reduce the hazards of depressive symptoms on cognitive function, if the elderly sleep enough (7–8 h for adults). Therefore, an enhanced focus on sleep duration and quality may be an important way forward for people exposed to alcohol and depressive symptoms.

However, this study also has certain limitations that need to be acknowledged. First, we cannot prove causation because this was an observational retrospective study. Those with neurocognitive disorders may be unable to accurately express their alcohol exposure, depressive symptoms, and sleep timing. Second, there may be selection bias due to some missing items, although participants with incomplete baseline information were excluded from the screening process. Third, similar to most previous studies [48,49], future studies need to provide objective assessments of sleep duration through repeated sampling, to address the abovementioned issue, rather than relying solely on retrospective sleep duration self-reports. However, this method is efficient and cost-effective in collecting data. Fourth, our study did not examine the multiplicative and additive interaction of alcohol consumption, sleep duration, and depressive symptoms on cognitive function. Fifth, there are many sensitive and specific alcohol markers closely related to alcohol consumption, such as liver tests such as aminotransferases (AST and ALT), gamma-glutamyl transpeptidase (GGT), carbohydrate-deficient transferrin (CDT), phosphatidyl ethanol (PEth), etc., to assess excessive alcohol consumption [50]. However, this study was limited and did not detect these serum markers, so exposure assessment was influenced due to this condition. Sixth, several cognitive assessments such as the Consortium to Establish a Registry for Alzheimer’s Disease (CERAD), developed for the diagnosis of cognitive dysfunction in Alzheimer’s disease, are utilized to examine cognitive decline. Such assessments are lacking in this study, so the assessment of cognitive function may not be accurate enough.

## 6. Conclusions

Depressive symptoms majorly mediate the relationship between alcohol consumption and cognitive function, and sleep duration changes the mediation effect in this model. Additionally, there is significant interaction between alcohol consumption, depressive symptoms, and sleep duration, and these factors are significantly associated with cognitive function. This may reduce the hazards of depressive symptoms on cognitive function, if the elderly sleep enough (7–8 h for adults). Hopefully, these findings will have implications for the development of alcohol policy-making and the prevention of cognitive decline, by reducing depressive symptoms and adjusting sleep duration.

## Figures and Tables

**Figure 1 ijerph-19-12574-f001:**
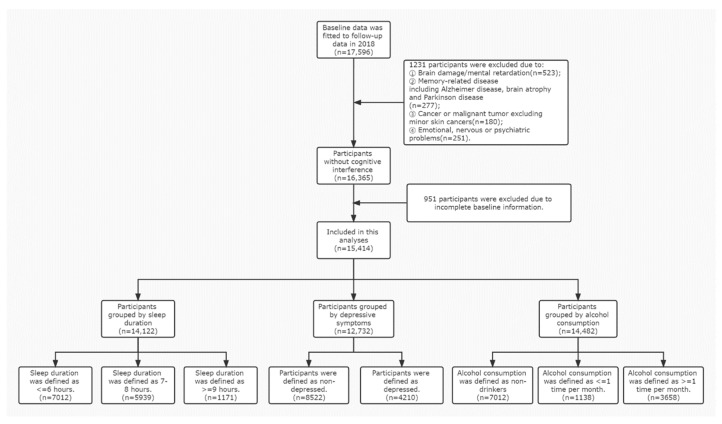
The process of participants’ selection and classification.

**Figure 2 ijerph-19-12574-f002:**
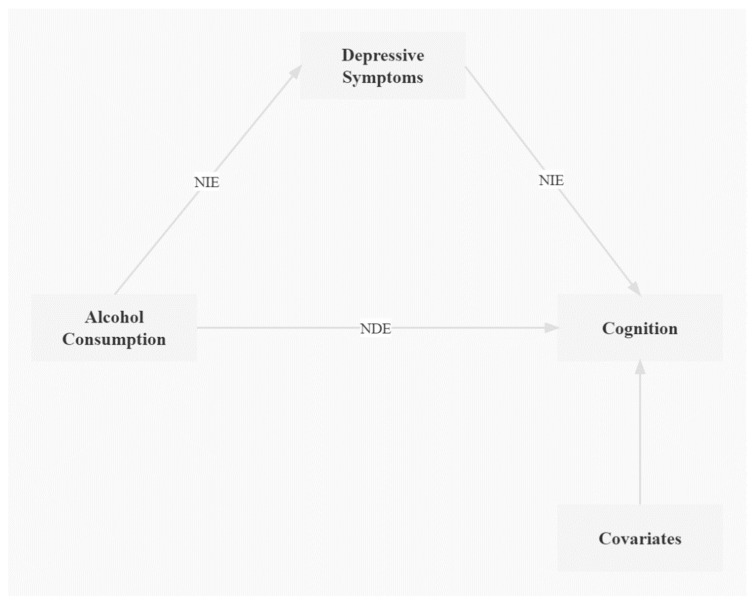
The model of mediation via depressive symptoms. Abbreviations: NIE, natural indirect effect; NDE, natural direct effect.

**Figure 3 ijerph-19-12574-f003:**
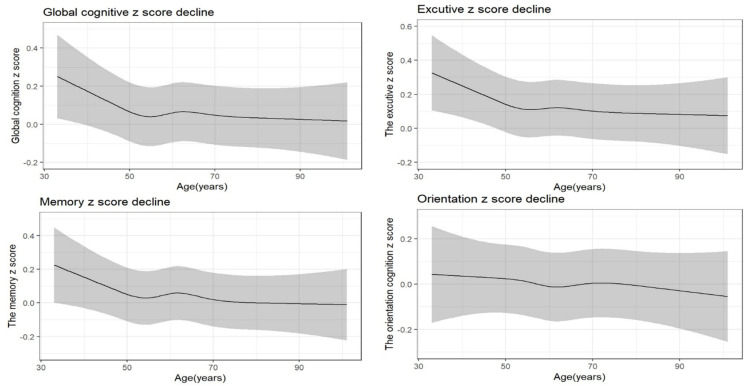
The trajectory of late-life cognitive decline with age.

**Figure 4 ijerph-19-12574-f004:**
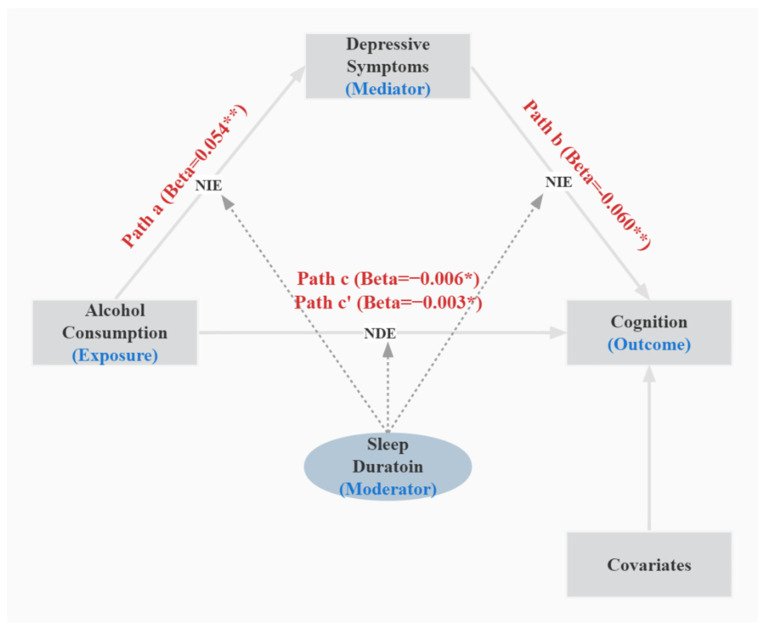
The model of conditional indirect effect via depressive symptoms when sleep duration was used as a moderator. After adjusting for gender, age, body mass index, blood pressure, education level, marriage, cigarette consumption, self-reported hypertension, diabetes, dyslipidemia, coronary heart disease, stroke, chronic lung disease, and asthma. Abbreviations: NIE, natural indirect effect; NDE, natural direct effect. Path a = exposure predicting mediator; path b = mediator predicting outcome; path c = exposure predicting outcome; path c’ = exposure predicting outcome. Beta = standardized regression coefficient. * *p* < 0.05 ** *p* < 0.01.

**Table 1 ijerph-19-12574-t001:** Baseline characteristics and differences between groups of the study population (totals and stratified according to alcohol consumption, depressive symptoms, and sleep duration).

	All	Alcohol Consumption (Times per Month) (n = 14,482)				Depressive Symptoms (n = 12,732)			Sleep Duration (Hours per Night) (n = 14,122)			
		Nondrinkers	≤1	>1	*p* Value	Nondepressed	Depressed	*p* Value	≤6 h	7–8 h	≥9 h	*p* Value
**N (%)**	15,414	9686 (66.9)	1138 (7.8)	3658 (25.3)		8522 (66.9)	4210 (33.1)		7012 (49.6)	5939 (42.1)	1171 (8.3)	
Age (years), mean (SD)	59.02 (10.14)	58.93 (10.10)	59.14 (9.92)	59.13 (10.24)	**0.532**	58.95 (10.12)	59.02 (10.09)	**0.691**	59.05 (10.14)	59.09 (10.20)	58.88 (9.75)	**0.813**
Gender (women), n (%)	8036 (52.1)	5027 (51.9)	590 (51.8)	1927 (52.7)	**0.698**	4396 (51.6)	2218 (52.7)	**0.237**	3669 (52.4)	3042 (51.2)	628 (53.7)	**0.218**
Education level, n (%)					**0.122**			**0.910**				**0.033**
No formal education or illiterate	4195 (27.3)	2570 (26.6)	311 (27.4)	1065 (29.2)		2287 (26.9)	1165 (27.8)		1909 (27.3)	1562 (26.4)	360 (30.9)	
Did not finish primary school	2676 (17.4)	1675 (17.4)	207 (18.2)	636 (17.5)		1466 (17.3)	716 (17.1)		1243 (17.8)	995 (16.8)	204 (17.5)	
Finished middle school	8062 (52.5)	5129 (53.1)	587 (51.7)	1856 (50.9)		4497 (53.0)	2197 (52.3)		3642 (52.1)	3192 (53.9)	576 (49.4)	
Higher and further education	368 (2.4)	241 (2.5)	27 (2.4)	72 (2.0)		209 (2.5)	102 (2.4)		173 (2.5)	146 (2.5)	20 (1.7)	
Others	63 (0.4)	39 (0.4)	4 (0.4)	15 (0.4)		33 (0.4)	17 (0.4)		24 (0.3)	25 (0.4)	5 (0.4)	
Living with a partner, n (%)	12,342 (80.1)	7784 (80.5)	906 (79.6)	2914 (79.8)	**0.016**	6859 (80.6)	3362 (80.0)	**0.516**	5635 (80.5)	4746 (80.1)	928 (79.5)	**0.598**
Cigarette consumption, n (%)	6052 (39.3)	3778 (39.1)	456 (40.2)	1470 (40.3)	**0.397**	3435 (40.4)	1601 (38.1)	**0.016**	2690(38.4)	2351 (39.6)	428 (36.5)	**0.098**
Blood pressure, mean (SD), mm Hg												
**Systolic**	**130.70 (21.67)**	**130.9 (21.83)**	**131.80 (22.72)**	**130.21 (21.24)**	**0.106**	**130.50 (21.83)**	**131.15 (21.47)**	**0.143**	**130.46 (21.56)**	**130.99 (21.84)**	**130.96 (21.75)**	**0.413**
Diastolic	75.95 (12.22)	76.07 (12.24)	76.34 (13.25)	75.67 (11.98)	**0.194**	75.82 (12.23)	76.18 (12.24)	**0.154**	75.91 (12.21)	75.99 (12.31)	75.91 (12.06)	**0.935**
Body mass index (kg/m^2^)	23.45 (3.95)	23.41 (3.95)	23.46 (3.68)	23.57 (4.08)	**0.185**	23.45 (4.05)	23.43 (3.85)	**0.730**	23.49 (4.02)	23.40 (3.87)	23.44 (4.03)	**0.516**
Cognitive function scores (z-scores, mean (SD))												
Memory score	−0.00 (1.00)	−0.03 (1.01)	0.16 (0.98)	0.03 (0.97)	**<0.001**	0.12 (1.01)	−0.18 (0.94)	**<0.001**	−0.02 (1.00)	0.03 (1.00)	−0.02 (1.00)	**0.038**
Orientation score	0.00 (1.00)	−0.04 (1.02)	0.10 (0.95)	0.08 (0.95)	**<0.001**	0.12 (0.95)	−0.19 (1.06)	**<0.001**	−0.01 (1.01)	0.04 (0.98)	−0.05 (1.02)	**0.007**
Executive score	−0.00 (1.03)	−0.08 (1.02)	0.12 (1.02)	0.17 (1.01)	**<0.001**	0.27 (0.94)	−0.09 (0.97)	**<0.001**	−0.02 (1.03)	0.02 (1.02)	−0.07 (1.04)	**0.009**
Global z scores	0.00 (1.00)	−0.07 (1.02)	0.19 (0.95)	0.12 (0.94)	**<0.001**	0.15 (0.97)	−0.26 (0.97)	**<0.001**	−0.002 (1.00)	0.03 (0.99)	−0.03 (1.00)	**0.017**
Hypertension n (%)	3712 (24.1)	2338 (24.3)	285 (25.2)	870 (24.0)	**0.725**	2101 (24.8)	984 (23.6)	**0.126**	1798 (25.8)	1341 (22.7)	286 (24.5)	**<0.001**
Diabetes, n (%)	847 (5.5)	545 (5.7)	73 (6.5)	181 (5.0)	**0.112**	470 (5.6)	226 (5.4)	**0.711**	422 (6.1)	319 (5.4)	50 (4.3)	**0.030**
Dyslipidemia, n (%)	1382 (9.0)	879 (9.3)	110 (9.8)	323 (9.0)	**0.712**	784 (9.4)	356 (8.6)	**0.156**	673 (9.8)	533 (9.1)	90 (7.8)	**0.075**
Coronary heart disease, n (%)	1800 (11.7)	1128 (11.7)	436 (12.0)	130 (11.6)	**0.885**	990 (11.7)	500 (12.0)	**0.650**	972 (14.0)	608 (10.3)	112 (9.6)	**<0.001**
Stroke, n (%)	291 (1.9)	187 (1.9)	15 (1.3)	74 (2.0)	**0.305**	164 (1.9)	79 (1.9)	**0.885**	146 (2.1)	92 (1.6)	24 (2.1)	**0.071**
Chronic lung disease, n (%)	1489 (9.7)	912 (9.5)	122 (10.8)	359 (9.9)	**0.342**	800 (9.4)	439 (10.5)	**0.063**	822 (11.8)	464 (7.8)	103 (8.8)	**<0.001**
Asthma, n (%)	515 (3.3)	320 (3.3)	38 (3.4)	126 (3.5)	**0.917**	294 (3.5)	140 (3.3)	**0.731**	286 (4.1)	150 (2.5)	40 (3.4)	**<0.001**

**Table 2 ijerph-19-12574-t002:** Cross-sectional association between sleep duration per night (hours per month) for depressive symptoms, alcohol consumption (times per month), covariates, and global cognitive function at baseline using generalized linear model.

	β, 95% CI							
	Memory Score		Orientation Score		Executive Score		Global Cognition z Score	
	Model 1 ^a^ (n = 10,335)	Model 2 ^b^ (n = 8215)	Model 1 ^a^ (n = 10,782)	Model 2 ^b^ (n = 8615)	Model 1 ^a^ (n = 11,580)	Model 2 ^b^ (n = 9257)	Model 1 ^a^ (n = 9710)	Model 2 ^b^ (n = 7715)
Alcohol Consumption (times per month)								
Nondrinkers	REF	REF	REF	REF	REF	REF	REF	REF
≤1	0.173	0.186	0.036	0.087	0.268	−0.070	0.204	0.193
	−0.006 to 0.352	−0.192 to 0.564	−0.108 to 0.180	−0.261 to 0.435	0.135 to 0.400	−0.393 to 0.253	0.055 to 0.352	−0.176 to 0.561
***p* value**	**0.058**	**0.334**	**0.623**	**0.624**	**<0.001**	**0.671**	**0.007**	**0.306**
>1	0.038	0.221	0.123	0.079	0.261	0.303	0.123	0.334
	−0.081 to 0.158	0.006 to 0.437	0.029 to 0.217	−0.127 to 0.286	0.175 to 0.347	0.110 to 0.496	0.026 to 0.220	0.120 to 0.548
***p* value**	**0.532**	**0.044**	**0.010**	**0.452**	**<0.001**	**0.002**	**0.013**	**0.002**
Depressive Symptoms								
Nondepressed	REF	REF	REF	REF	REF	REF	REF	REF
Depressed	−0.368	−0.411	−0.223	−0.367	−0.337	−0.464	−0.396	−0.496
	−0.496 to −0.239	−0.604 to −0.219	−0.319 to −0.128	−0.553 to −0.182	−0.426 to −0.248	−0.632 to−0.296	−0.494 to −0.298	−0.694 to −0.297
***p* value**	**<0.001**	**<0.001**	**<0.001**	**<0.001**	**<0.001**	**<0.001**	**<0.001**	**<0.001**
Sleep Duration (hours per night)								
≤6 h	−0.148	0.006	−0.053	−0.084	−0.082	−0.055	−0.105	−0.025
	−0.240 to −0.056	−0.166 to 0.177	−0.133 to 0.028	−0.252 to 0.085	−0.157 to −0.006	−0.210 to 0.101	−0.187 to −0.023	−0.198 to 0.147
***p* value**	**0.002**	**0.947**	**0.199**	**0.331**	**0.033**	**0.491**	**0.012**	**0.772**
7–8 h	REF	REF	REF	REF	REF	REF	REF	REF
≥9 h	0.047	0.128	−0.074	−0.070	0.003	0.218	0.079	0.158
	−0.124 to 0.217	−0.218 to 0.474	−0.222 to 0.074	−0.389 to 0.249	−0.135 to 0.142	−0.082 to 0.518	−0.075 to 0.232	−0.185 to 0.501
***p* value**	**0.592**	**0.468**	**0.326**	**0.667**	**0.962**	**0.154**	**0.315**	**0.367**
Age (years)	0.000	−0.001	0.000	−0.001	0.001	0.001	0.000	−0.001
	−0.002 to 0.002	−0.004 to 0.001	−0.002 to 0.002	−0.03 to 0.001	−0.001 to 0.003	−0.001 to 0.003	−0.002 to 0.002	−0.003 to 0.002
***p* value**	**0.939**	**0.291**	**0.839**	**0.448**	**0.219**	**0.457**	**0.747**	**0.654**
Gender	0.014	0.011	0.023	0.018	0.010	−0.010	0.012	0.006
	−0.026 to 0.054	−0.035 to 0.057	−0.016 to 0.063	−0.026 to 0.063	−0.027 to 0.046	−0.051 to 0.031	−0.029 to 0.053	−0.040 to 0.051
***p* value**	**0.497**	**0.634**	**0.245**	**0.417**	**0.600**	**0.626**	**0.558**	**0.811**
Education Level	0.011	0.007	0.016	0.012	0.016	0.011	0.018	0.016
	0.000 to 0.021	−0.005 to 0.019	0.006 to 0.026	0.001 to 0.024	0.007 to 0.025	0.001 to 0.022	0.007 to 0.028	0.004 to 0.028
***p* value**	**0.042**	**0.258**	**0.002**	**0.037**	**0.001**	**0.035**	**0.001**	**0.009**
Marriage	−0.002	0.001	−0.008	−0.002	0.002	0.007	0.002	0.007
	−0.017 to 0.014	−0.016 to 0.018	−0.023 to 0.007	−0.019 to 0.014	−0.011 to 0.016	−0.008 to 0.022	−0.013 to 0.017	−0.011 to 0.024
***p* value**	**0.838**	**0.906**	**0.302**	**0.790**	**0.736**	**0.356**	**0.794**	**0.452**
Blood Pressure								
Systolic	NA	0.001	NA	0.001	NA	0.000	NA	0.001
		−0.001 to 0.002		0.000 to 0.002		−0.001 to 0.002		−0.001 to 0.002
***p* value**		**0.463**		**0.110**		**0.512**		**0.315**
Diastolic	NA	−0.001	NA	−0.003	NA	−0.003	NA	−0.002
		−0.004 to 0.002		−0.005 to −0.001		−0.005 to 0.000		−0.004 to 0.001
***p* value**		**0.416**		**0.018**		**0.022**		**0.226**
Body Mass Index (kg/m^2^)	NA	0.000	NA	−0.001	NA	−0.002	NA	−0.002
		−0.006 to 0.005		−0.006 to 0.004		−0.007 to 0.002		−0.008 to 0.003
***p* value**		**0.903**		**0.733**		**0.325**		**0.397**
Cigarette Consumption	NA	0.025	NA	0.043	NA	0.025	NA	0.048
		−0.019 to 0.069		0.000 to 0.086		−0.015 to 0.064		0.004 to 0.092
***p* value**		**0.267**		**0.048**		**0.216**		**0.034**
Hypertension	NA	0.048	NA	−0.006	NA	−0.016	NA	0.014
		−0.005 to 0.101		−0.057 to 0.046		−0.063 to 0.032		−0.040 to 0.067
***p* value**		**0.076**		**0.829**		**0.515**		**0.617**
Diabetes	NA	−0.061	NA	−0.026	NA	−0.013	NA	−0.074
		−0.160 to 0.039		−0.122 to 0.071		−0.101 to 0.076		−0.175 to 0.026
***p* value**		**0.233**		**0.602**		**0.780**		**0.145**
Dyslipidemia	NA	0.044	NA	0.031	NA	−0.008	NA	0.050
		−0.036 to 0.125		−0.047 to 0.110		−0.080 to 0.064		−0.030 to 0.131
***p* value**		**0.278**		**0.434**		**0.834**		**0.222**
Coronary Heart Disease	NA	−0.059	NA	0.034	NA	0.004	NA	−0.054
		−0.127 to 0.010		−0.033 to 0.102		−0.058 to 0.065		−0.124 to 0.015
***p* value**		**0.093**		**0.315**		**0.904**		**0.124**
Stroke	NA	−0.087	NA	0.049	NA	0.028	NA	−0.003
		−0.245 to 0.072		−0.106 to 0.204		−0.114 to 0.170		−0.163 to 0.157
***p* value**		**0.284**		**0.535**		**0.704**		**0.971**
Chronic Lung Disease	NA	0.003	NA	−0.006	NA	0.031	NA	0.002
		−0.072 to 0.078		−0.079 to 0.068		−0.036 to 0.098		−0.074 to 0.078
***p* value**		**0.943**		**0.879**		**0.370**		**0.951**
Asthma	NA	0.156	NA	0.051	NA	0.029	NA	0.095
		0.032 to 0.279		−0.068 to 0.170		−0.080 to 0.139		−0.029 to 0.219
***p* value**		**0.013**		**0.403**		**0.601**		**0.133**

^a^ Model 1 was adjusted for age, gender, education level, and marital status. ^b^ Model 2 was the same as model 1 plus systolic, diastolic, BMI, cigarette consumption, self-reported hypertension, diabetes, dyslipidemia, coronary heart disease, stroke, chronic lung disease, and asthma. Abbreviations: NA, not applicable; REF, reference.

**Table 3 ijerph-19-12574-t003:** Mean difference in overall rate of change in cognitive decline during follow-up using generalized estimating equation.

	β (95% CI) ^a^	*p* Value
Sleep Duration (hours per night)		
≤6 h	−0.056 (NA)	<0.001
7–8 h	REF	REF
≥9 h	0.225 (NA)	<0.001
Depressive Symptoms		
Nondepressed	REF	REF
Depressed	−0.271 (−0.411 to −0.130)	<0.001
Alcohol Consumption (times per month)		
No	REF	REF
≤1	0.072 (−0.184 to 0.327)	0.582
>1	0.156 (0.007 to 0.305)	<0.05
Depressive Symptoms * Alcohol Consumption		0.539
Sleep Duration * Alcohol Consumption		0.305
Depressive Symptoms * Sleep Duration		0.324
Depressive Symptoms * Alcohol Consumption * Sleep Duration		<0.05
Age (years)	−0.001 (−0.003 to 0.000)	0.114
Gender	0.034 (0.002 to 0.067)	<0.05
Education Level	0.013 (0.005 to 0.022)	<0.05
Marriage	0.007 (−0.005 to 0.020)	0.252
Blood Pressure		
Systolic	0.001 (0.000 to 0.002)	0.164
Diastolic	0.000 (−0.002 to 0.001)	0.637
Body Mass Index (kg/m^2^)	−0.001 (−0.004 to 0.003)	0.793
Cigarette Consumption	0.036 (0.004 to 0.068)	<0.05
Hypertension	0.019 (−0.019 to 0.057)	0.327
Diabetes	−0.060 (−0.132 to 0.011)	0.097
Dyslipidemia	−0.017 (−0.075 to 0.042)	0.577
Coronary Heart Disease	−0.008 (−0.057 to 0.042)	0.767
Stroke	NA (−0.116 to 0.116)	1.000
Chronic Lung Disease	0.026 (−0.028 to 0.079)	0.349
Asthma	0.026(−0.061 to 0.113)	0.557

^a^ After adjusting for gender, age, body mass index, blood pressure, education level, marriage, cigarette consumption, self-reported hypertension, diabetes, dyslipidemia, coronary heart disease, stroke, chronic lung disease, and asthma. Abbreviations: NA, not applicable. *, interaction.

**Table 4 ijerph-19-12574-t004:** Adjusted direct and indirect associations of alcohol consumption with global cognitive function mediated via depressive symptoms.

	Model 1 ^a^ (n= 11,416)		Model 2 ^b^ (n = 11,344)		Model 3 ^c^ (n = 9000)	
	β(95% CI)	*p* Value	β(95% CI)	*p* Value	β(95% CI)	*p* Value
Alcohol Consumption						
Total Association	−0.008 (−0.030 to 0.013)	**0.447**	−0.009 (−0.031 to 0.012)	**0.404**	−0.006 (−0.031 to 0.018)	**0.600**
Direct Association	−0.005 (−0.027 to 0.016)	**0.635**	−0.006 (−0.027 to 0.016)	**0.585**	−0.003 (−0.027 to 0.021)	**0.793**
Indirect Association via Depressive Symptoms	−0.003 (−0.005 to −0.001)	**0.001**	−0.003 (−0.005 to −0.001)	**0.001**	−0.003 (−0.005 to −0.001)	**0.003**
Proportion Mediated, %	100		100		100	

^a^ Model 1 was a crude model. ^b^ Model 2 was adjusted for age, gender, education level, and marital status. ^c^ Model 3 was the same as model 1 plus systolic, diastolic, BMI, and cigarette consumption.

**Table 5 ijerph-19-12574-t005:** The results of conditional indirect effect via depressive symptoms when sleep duration was used as a moderator.

		Model 1 ^a^ (n = 10,460)		Model 2 ^b^ (n = 10,396)		Model 3 ^c^ (n = 8300)	
		Effect	95% CI	Effect	95% CI	Effect	95% CI
Depressive Symptoms	Low level (−1 SD)	−0.004	−0.008 to −0.001	−0.004	−0.008 to −0.001	−0.004	−0.008 to −0.001
	Mean level	−0.004	−0.005 to −0.001	−0.003	−0.006 to −0.001	−0.003	−0.005 to −0.001
	High level (+1 SD)	−0.003	−0.005 to 0.001	−0.002	−0.006 to 0.000	−0.002	−0.005 to 0.001

^a^ Model 1 was a crude model. ^b^ Model 2 was adjusted for age, gender, education level, and marital status. ^c^ Model 3 was the same as model 1 plus systolic, diastolic, BMI, and cigarette consumption.

## Data Availability

The data are available at http://charls.pku.edu.cn (accessed on 20 December 2021).

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
