# Peer review of "The Influence of Alcohol Consumption, Depressive Symptoms and Sleep Duration on Cognition: Results from the China Health and Retirement Longitudinal Study"

_ijerph, 2022, doi:10.3390/ijerph191912574_

Round 1
Reviewer 1 Report
This is recomended to organise the whole article using the related Equator Checklist from: The EQUATOR Network | Enhancing the QUAlity and Transparency Of Health Research (equator-network.org)
Please fill out the checklist that belongs to your article methods and attach it to your article as supplementary file.
Abstract
What is the exact resrearch method? Is this a cross-sectional study?
The research setting and country should be identified also.
Introduction
It should contain a brief description of previous studies conducted on this topic and what has been the gap of knowledge leading to this study.
Methods
Please elaborate on the cohort groups and how they hav been classified and why.
Abbreviations under the figure should be described.
Describe any ethical considerations taken for this research.
Conclusion
You need to indicate the practical implications of this study for education, research and policy making.
Reviewer 2 Report
This is an interesting study with many thousand participants. However, the study and its results are difficult to understand. This is potentially due to the unusual use of the English language.
Title: The title should be easier to understand, for example: "The influence of alcohol consumption, depressive symptoms and sleep duration on cognition: Results from the China Health and Retirement Longitudinal Study"
Abstract: All abbreviations must be explained when they are used for the first time, even in the abstract. The results section in the abstract is difficult to follow. I suggest to write first whether alcohol consumption, depressive symptoms and sleep duration on their own had an influence of cognition or cognitive decline. Afterwards, the authors can report whether one of these factors mediated the effect of another factor on cognition. This approach has been nicely explained by the authors in the last paragraph of the introduction. However, in the abstract, the reader cannot understand what the authors did. They write a lot about mediation etc., when the reader should comprehend the basic structure of the study design first.
Introduction: The content of the introduction is good, but the use of the English language is partly unusual. The manuscript would benefit from a native speaker as an editor or co-author.
Methods: All abbreviations in a figure or table should be explained in the legend. Please explain natural indirect effect (NIE) and natural direct effect (NDE) in the legend of figure 2.
Was alcohol consumption measured or just asked? Did the authors measure a biological marker of alcohol consumption like GGT, MCV or CDT in the serum? If not, this should be mentioned in the discussion.
Results: The arrows in Figure 4 do not indicate whether the relations are positive and favorable or negative. It should be clear for the reader that alcohol has a positive effect on cognition, whereas depressive symptoms have a negative effect on cognition.
Discussion: The methodological shortcomings of the study should be mentioned in more detail. The authors did not use a standardized neuropsychological test battery like the CERAD. This should be stated. And there was no objective control like a biological marker of alcohol consumption to check whether the participants' answers around alcohol were true. Quite often, people deny the amount of alcohol they consume.
Round 2
Reviewer 1 Report
Nothing more.